# Expanding Horizons: Host Range Evolution and Treatment Strategies for Highly Pathogenic Avian Influenza H5N1 and H7N9

**DOI:** 10.3390/v18010054

**Published:** 2025-12-30

**Authors:** Nika Heidari Gazik, Mark Holodniy, Vafa Bayat

**Affiliations:** 1Bahá’í Institute for Higher Education (BIHE), Tehran 11369, Iran; nika.heidarigazik@postgrad.manchester.ac.uk; 2University of Manchester, Manchester M139PL, UK; 3Division of Infectious Diseases and Geographic Medicine, Stanford University, Stanford, CA 90024, USA; mark.holodniy@va.gov; 4VHA Pathology & Laboratory Medicine Service, Washington, DC 20005, USA; 5Bitscopic Inc., Dover, DE 19901, USA

**Keywords:** avian influenza virus, H5N1, H7N9, influenza epidemiology, mammal-to-human transmission, viral evolution

## Abstract

Avian influenza viruses (AIVs), including H5N1 and H7N9, from the Orthomyxoviridae family present substantial public health concerns. The predominant circulating clade 2.3.4.4b has demonstrated enhanced capacity for mammalian adaptation, raising concerns about potential reassortment with human seasonal influenza viruses. Unlike H7N9’s limited host range, H5N1 infects birds, various mammals, and humans. Recent concerns include widespread H5N1 infection of U.S. dairy cattle across 18 states, affecting over 1000 herds with 71 human infections (70 H5N1 and 1 H5N5). Key observations include cow-to-cow transmission, viral presence in milk, and transmission to humans, mainly through occupational exposure. Evidence of mammal-to-mammal transmission has been documented in European and Canadian foxes and South American marine mammals. Standard pasteurization effectively inactivates the virus in milk. The continuing mammalian adaptations, particularly mutations like PB2-E627K, PB2-D701N, and PB2-M535I, suggest potential for further evolution in new hosts, emphasizing the need for enhanced surveillance to mitigate pandemic risks.

## 1. Introduction

Avian influenza viruses (AIVs) are negative-sense, single-stranded RNA viruses belonging to the Orthomyxoviridae family, genus influenza A virus. The AIV genome consists of eight RNA segments encoding 11 proteins [1,2]. Two glycoproteins on the viral envelope, hemagglutinin (HA) and neuraminidase (NA), are critical determinants of AIV pathogenicity and host tropism. Based on the antigenic properties of these surface proteins, AIVs are classified into subtypes H1–H18 and N1–N11 [1].

The H5 and H7 subtypes are of particular concern due to their potential to cause severe disease and mortality in humans and poultry. H5N1 and H7N9 are two notable strains that have caused human infections and fatalities. The first human cases of H5N1 and H7N9 were reported in Hong Kong in 1997 and China in 2013, respectively [3]. As of May 2025, it is estimated that H5N1 has infected 973 people worldwide with a case-fatality rate of 48%, while H7N9 has infected over 1500 people with a 40% case-fatality rate [4,5].

A key difference between these two strains is their host range. H7N9 primarily infects poultry and humans, with limited mammalian infections reported mainly in laboratory settings [6,7]. In contrast, H5N1 has a broad host range that includes wild birds, domestic poultry, and various mammalian species such as cats, dogs, foxes, mink, seals, whales, and pigs [8,9]. This ability to infect a wide range of hosts, coupled with the acquisition of mammalian-adapting mutations, raises concerns about the pandemic potential of H5N1 [10].

Recent years have seen an alarming increase in H5N1 infections among mammals worldwide, with a notable surge in cases across the Americas [11]. Of particular concern is the 2024 outbreak of H5N1 in U.S. dairy cattle, a species typically considered resistant to influenza A viruses [12]. In March 2024, HPAI (Highly pathogenic avian influenza) H5N1 was detected in dairy cattle in the United States, marking the first time this virus was found in cows [13]. By late 2024, the virus had spread to over 1063 herds across 18 states, with 70 human infections related to dairy cow exposure, 24 cases linked to occupational exposure during culling of H5N1-infected poultry, 2 from other animal exposure, and 3 with undetermined exposure. This outbreak has featured cow-to-cow transmission, contamination of milk with H5N1, and subsequent human infections primarily through occupational exposure to contaminated milk or direct contact with infected cows, often resulting in conjunctivitis [14,15].

The most dramatic evidence of H5N1’s expanding mammalian host range occurred during 2023 in South America, where unprecedented mass mortality events affected South American sea lions (Otaria flavescens) along the Pacific and Atlantic coasts. At least 24,000 sea lions died across five South American countries during this outbreak period [16], representing the largest documented H5N1 mortality event in marine mammals to date. This unprecedented scale demonstrates the virus’s capacity for sustained transmission in mammalian populations beyond traditional avian hosts.

In Peru, surveillance documented 5224 sea lion deaths between January and April 2023, representing approximately 5% of the Peruvian population of this species [16]. Clinical presentations included predominantly neurological signs such as tremors, convulsions, and paralysis, alongside respiratory manifestations including dyspnea and tachypnea. Chile reported even higher proportional mortality, with 4545 sea lions stranded and deceased by June 2023, representing approximately 12% of their national population [17]. Necropsy findings revealed acute encephalitis and interstitial pneumonia consistent with systemic HPAI H5N1 infection, with molecular analysis confirming HPAI H5N1 clade 2.3.4.4b and 20% of sampled animals testing positive for influenza A virus.

The outbreak subsequently spread to Argentina, Uruguay, and Brazil between August and October 2023, with genetic analysis suggesting that the virus traveled approximately 5000 km from Peru to Tierra del Fuego via the Pacific Ocean, then 2800 km northward along the Atlantic coast. The most probable transmission route was spillover from infected wild birds, particularly during shared feeding activities, although potential sea lion-to-sea lion transmission cannot be excluded, given the gregarious colonial breeding behavior of this species. Pathological examination demonstrated severe non-suppurative meningoencephalitis with neuronal necrosis and viral antigen detected in brain tissue by immunohistochemistry, with detection of mammalian-adaptive mutations including PB2-Q591K and PB2-D701N in viral genomes from multiple countries [18].

Additionally, the January 2023 human case in Ecuador, involving a 9-year-old girl with severe pneumonia who had contact with backyard poultry, underscores the continued risk of zoonotic transmission in regions with endemic circulation of HPAI H5N1 in wild bird populations [19].

During September–December 2024, California reported 38 human cases of HPAI A(H5N1), with 36 being dairy farm workers with occupational exposure and two with undetermined exposure—the first pediatric HPAI A(H5N1) case reported in the United States [20]. Most patients experienced mild illness [21]; however, in January 2025, Louisiana reported the first U.S. death from H5N1, involving a patient over 65 years with underlying medical conditions who was exposed to sick and dead birds from a backyard flock [22], which suggests that while zoonotic transmission is occurring, the current strain may cause less severe disease in humans than historical H5N1 strains.

Other concerning developments include evidence of mammal-to-mammal transmission in foxes, seals, and tigers [23,24], as well as foodborne transmission and neurological manifestations in various mammalian species [25,26].

In light of these events, this review aims to summarize the current knowledge on the virology, epidemiology, and evolution of AIVs, with a focus on the expanding host range and pandemic potential of H5N1. Understanding the viral factors and host–pathogen interactions driving the increased mammalian susceptibility to H5N1 will be crucial for developing effective surveillance strategies and countermeasures to mitigate the risk of a future pandemic.

## 2. Virology of Avian Influenza Viruses

### 2.1. Viral Genome and Structure

AIVs are enveloped, pleomorphic viruses with a spherical shape, ranging from 80 to 120 nm in size. The virus consists of three main subviral components: an envelope made of a lipid bilayer membrane, a layer of matrix 1 proteins (M1), and a viral ribonucleoprotein (vRNP) core. The viral genome comprises eight negative-sense, single-stranded RNA segments that encode 11 essential proteins, including the surface glycoproteins hemagglutinin (HA) and neuraminidase (NA), the polymerase complex proteins (PB2 (polymerase basic 2), PB1, and PA (polymerase acidic), the nucleoprotein (NP), the matrix proteins (M1 and M2), and the non-structural proteins (NS1 and NS2/NEP) [1,2].

The viral envelope, derived from the host cell membrane, contains spikes formed by two integral transmembrane glycoproteins (HA and NA) and a transmembrane ion channel matrix 2 (M2). These glycoproteins play crucial roles in viral entry and release: HA binds to sialic acid receptors on the host cell surface and mediates membrane fusion, while NA cleaves sialic acid residues from glycoproteins and glycolipids, facilitating the release of newly formed virions from infected cells [1]. The polymerase complex (PB2, PB1, and PA) and non-structural proteins (NS1 and NS2/NEP) are critical for viral replication and host adaptation, particularly in strains like H5N1 and H7N9. For instance, adaptive mutations in PB2, such as the E627K substitution, enhance polymerase activity in mammalian hosts, increasing the zoonotic potential of H5N1, while NS1 mutations (e.g., deletions at positions 80–84 or P42S) modulate host immune responses by antagonizing interferon production, contributing to the pathogenicity of both H5N1 and H7N9 [1,3].

AIVs are classified into subtypes based on the antigenic properties of their HA and NA proteins. To date, 18 HA subtypes (H1–H18) and 11 NA subtypes (N1–N11) have been identified, with various combinations observed in nature, including the highly pathogenic H5N1 and H7N9 strains that pose significant public health risks [1].

### 2.2. Pathogenicity and Host Range

AIVs can be further categorized as low-pathogenic avian influenza (LPAI) or highly pathogenic avian influenza (HPAI) viruses, based on their ability to cause disease in poultry [2]. LPAI viruses typically cause mild or subclinical infections in birds, while HPAI viruses can lead to severe, systemic disease with high mortality rates [27].

The pathogenicity of AIVs is largely determined by the cleavability of the HA protein, which dictates the virus’s ability to be activated by host proteases, with highly pathogenic strains possessing a polybasic cleavage site that allows cleavage by ubiquitous intracellular proteases like furin, enabling systemic spread and severe disease. LPAI viruses possess a specific configuration at the HA cleavage site: a single arginine residue with another basic arginine or lysine residue at position 3 or 4 from this site. This structure requires extracellular host proteases, specifically trypsin-like enzymes, which are found only in the respiratory and intestinal tracts of birds [28]. This restricted protease distribution naturally limits viral spread within the host.

In contrast, HPAI viruses have evolved multiple basic arginine and lysine amino acids at the HA cleavage site, either through insertion or substitution. This modification allows the virus to be cleaved by ubiquitous intracellular proteases, particularly furin, enabling viral replication across various vital organs and facilitating systemic spread [29]. This broader tissue tropism explains the increased virulence and mortality associated with HPAI viruses.

While AIVs primarily infect avian species, highly pathogenic subtypes, particularly H5 and H7 (e.g., H5N1 and H7N9), have demonstrated the ability to cross species barriers and infect mammals, including humans [6]. This zoonotic potential is influenced by various factors, such as the viral receptor binding specificity, polymerase activity, and host immune responses [30,31].

### 2.3. Viral Replication and Mutation

AIVs replicate in the nucleus of infected cells, utilizing the viral RNA-dependent RNA polymerase complex (RdRp) composed of PB2, PB1, and PA [32]. The error-prone nature of the RdRp leads to a high mutation rate, allowing AIVs to evolve rapidly and adapt to new hosts [6].

Two primary mechanisms drive the evolution of AIVs: antigenic drift and antigenic shift. Antigenic drift involves the accumulation of point mutations in the HA and NA genes, leading to minor changes in the antigenic properties of the virus [29]. Antigenic shift, a key mechanism of rapid AIV evolution, occurs when two or more subtypes co-infect a host cell and reassort gene segments, producing novel strains with pandemic potential that are shaped by selection pressures [6].

Understanding the virological characteristics of AIVs, including their genome organization, pathogenicity determinants, and evolutionary mechanisms, is essential for assessing the risks posed by emerging strains and developing effective prevention and control strategies. As the recent expansion of H5N1 into new mammalian hosts demonstrates, continued research into the factors driving AIV evolution and cross-species transmission is critical for mitigating the impact of potential pandemics.

## 3. Epidemiology and Evolution of AIVs

### 3.1. H7N9 Subtype

The H7N9 subtype first emerged in humans in China in 2013, causing severe respiratory illness and high mortality [6]. As of 2017, there have been five epidemic waves of H7N9 infection, with the fifth wave in 2016–2017 being the largest [33]. In total, over 1500 human cases have been reported, with a case-fatality rate of approximately 40% [34].

H7N9 primarily infects poultry and humans, with limited mammalian infections reported mainly in laboratory settings [6,7]. Transmission to humans occurs through direct or indirect exposure to infected poultry, particularly in live bird markets [3]. Although some cases of limited human-to-human transmission have been suspected, sustained transmission between humans has not been observed [6].

Several key mutations in the H7N9 genome have been identified as important for its adaptation to humans. These include changes in the hemagglutinin (HA) protein that enhance binding to human-type receptors (e.g., Q226L), as well as mutations in the PB2 protein (e.g., E627K, D701N) that increase viral replication efficiency in mammalian cells [6,34]. However, despite these adaptations, the H7N9 virus has not acquired the ability to transmit efficiently between humans, limiting its pandemic potential.

### 3.2. H5N1 Subtype

A key factor contributing to the pandemic potential of H5N1 is its broad host range. In addition to wild birds and domestic poultry, the virus has been found to infect various mammalian species, including cats, dogs [23,24], mink, seals, whales, and pigs [8]. This ability to cross species barriers and adapt to new hosts raises concerns about the virus’s evolutionary capacity.

Phylogenetic analyses have revealed that H5N1 has undergone significant genetic diversification, with multiple clades and subclades emerging over time [2]. Some of these lineages, such as clade 2.3.4.4, have been associated with increased transmissibility and virulence in mammals [30]. Key mutations that facilitate mammalian adaptation include changes in the HA protein that enhance binding to human-type receptors (e.g., S159N, S227N) and mutations in the PB2 protein that increase polymerase activity and viral replication in mammalian cells (e.g., E627K, D701N, and M535I) [30,31].

In recent years, the frequency and geographic range of H5N1 infections in mammals have expanded dramatically. Between 2020 and 2023, over 280 outbreaks in mammals were reported in the Americas alone, with skunks and foxes being among the most affected species [11]. Notably, human infections have also emerged, with the United States reporting its first H5N1 case in Colorado in 2022, followed by cases in Cambodia and Chile in 2023. Ecuador documented South America’s first human case that same year―a three-year-old girl infected with clade 2.3.4.4b [19].

The most concerning development has been the 2024 U.S. cattle-poultry outbreak. Between March 2024 and November 2025, 70 human H5N1 cases and 1 H5N5 case were reported [35] in the United States (41 exposed to dairy cows, 24 to commercial poultry, 2 to backyard poultry, and 3 with unknown exposure) [32]. These events, along with the increasing detection of mammalian-adapting mutations, highlight the need for heightened surveillance and preparedness efforts to mitigate the risk of further H5N1 evolution and potential human-to-human transmission.

## 4. Signs of H5N1 Pandemic Risk

### 4.1. Mammalian Adaptations and Neurotropism

The growing number of H5N1 infections in mammals, coupled with the identification of key mutations that facilitate mammalian adaptation, underscores the pandemic potential of this virus. Of particular concern are recent reports of mammal-to-mammal transmission, foodborne infections, and neurological manifestations in various species.

In foxes and other mesocarnivores, H5N1 has been found to exhibit significant neurotropism with extensive pathological findings. Detailed examinations have revealed viral antigens throughout all brain structures, with predominant involvement of neurons and neuropil. The pathology includes multifocal necrotic neurons and glial cells, neutrophilic infiltrates in the cortex, and perivascular cuffing. Additional findings include gliosis, hemorrhage, neuronophagia, meningitis, and vasculitis, demonstrating the virus’s capacity to cause severe and widespread neurological damage [20,22].

Marine mammals, particularly seals, have shown similarly concerning manifestations. Studies have documented meningoencephalitis in 100% of examined cases, accompanied by fibrinosuppurative alveolitis and multiorgan acute necrotizing inflammation. Virus antigens have been detected throughout multiple tissues, including neuropils and neurons, renal glomeruli, pulmonary alveolar septa, glandular bronchial cells, lymph nodes, spleen, pancreas, and liver [24,26]. Notably, these marine mammal cases have been associated with mammalian-adapting mutations in the PB2 protein, such as E627K, D701N, and M535I, similar to mutations observed in fox infections.

The ability of H5N1 to infect the central nervous system and cause such extensive neurological disease in mammals is a worrying development, as it suggests the virus may be evolving towards a more virulent phenotype. Moreover, the detection of mammal-to-mammal transmission in foxes, seals, tigers, and domestic cats [14,23,24] indicates that H5N1 is becoming increasingly adapted to mammalian hosts, potentially setting the stage for more efficient transmission in humans.

### 4.2. U.S. Cattle Outbreak and Milk Contamination

In March 2024, an outbreak of H5N1 in dairy cattle (not beef cattle) was reported in the United States, marking the first time this virus has been detected in cows [14]. To date, three distinct spillover events have been identified in U.S. dairy cattle, involving genotypes B3.13 and D1.1, highlighting the virus’s expanding host range [14,15,35]. This event is particularly concerning, as cattle are typically considered resistant to influenza A viruses [12].

Key observations from the outbreak include cow-to-cow transmission, contamination of milk with H5N1, and subsequent human infections [14,15]. To date, there have been no reported cases of beef cattle infected, and hypotheses remain untested [36]. Infected cows exhibited respiratory symptoms, reduced milk production, and neurological signs, while humans exposed to infected animals mainly developed conjunctivitis [12,37]. Notably, the H5N1 strain involved in this outbreak belongs to clade 2.3.4.4b and harbors mutations associated with mammalian adaptation, such as PB2-E627K, PB2-D701N, and PB2-M535I [30,31].

The U.S. cattle outbreak represents a significant development in the evolution of H5N1, as it demonstrates the virus’s ability to infect a new mammalian host, transmit efficiently within the species, and contaminate animal products consumed by humans. This event, along with the growing evidence of foodborne transmission in other mammals (e.g., foxes, seals), highlights the need for enhanced surveillance and biosecurity measures to prevent further spillover and potential human-to-human transmission.

## 5. Molecular Aspects and Virulence Enhancement

Recent molecular studies have provided critical insights into why current H5N1 strains are showing enhanced virulence and transmissibility in mammals. Comparative research found that a Texas dairy cattle H5N1 virus showed superior growth capability and rapid replication kinetics in human lung cell lines compared to older HPAI isolates. In vivo studies demonstrated that cattle H5N1 exhibited more intense pathogenicity in mice, with rapid lung pathology, high virus titers in the brain, and high mortality after challenge via different inoculation routes. The virus also demonstrated efficient antagonism of overexpressed retinoic acid-inducible gene I (RIG-I)- and melanoma differentiation-associated protein 5 (MDA5)-mediated innate antiviral signaling pathways [38].

Analysis of the HA protein from cattle-infecting H5N1 viruses revealed they have acquired mutations enabling slight binding to human-like 2-6-linked receptors while still exhibiting a strong preference for avian-like 2-3-linked sialic acid receptors. Immunohistochemical staining has shown that the HA protein binds to bovine pulmonary and mammary tissues, as well as human conjunctival, tracheal, and mammary tissues, indicating potential routes for human transmission. High-resolution cryo-electron microscopy (EM) structures of this H5 HA in complex with either receptor have elucidated the molecular mechanisms underlying its binding properties [39].

Surprisingly, studies of the receptor-binding site of H5N1 hemagglutinin revealed that it was already occupied by a 2-3-linked sialic acid that emanated from asparagine N169 of a neighboring protomer on hemagglutinin itself. This “auto-glycan recognition” may play a role in viral dispersal and adds to the complexities surrounding H5N1 zoonosis [40].

Importantly, H5N1 strains demonstrating dual affinity for both α2,3-linked (avian-type) and α2,6-linked (human-type) sialic acid receptors pose particular concern, as this dual tropism may facilitate co-infection with human seasonal influenza viruses within the same host cell, thereby increasing the risk of genetic reassortment and emergence of novel pandemic strains.

The role of phosphorylation in regulating viral proteins and enhancing viral fitness has also been studied. Research has shown that phosphorylation of the polymerase acidic (PA) protein at site S225 enhances the viral fitness of highly pathogenic H5N1 virus in mammals by assuring effective viral ribonucleoprotein complex activity. This modification enhances viral replication and virulence in mice, suggesting a molecular mechanism for adaptation to mammalian hosts [41].

## 6. Immune Response and Cross-Protection

Understanding the immune response to H5N1 viruses is crucial for assessing pandemic risk and developing effective countermeasures. Recent studies in ferrets have shown that prior exposure to seasonal influenza A(H1N1)pdm09 virus may provide some cross-protection against current H5N1 strains. Ferrets with previous H1N1pdm09 immunity showed differential tissue tropism, reduced viral dissemination outside the respiratory tract, and lower viral loads in nasal secretions and the respiratory tract compared to naive ferrets, when challenged with H5N1 [42]. However, this cross-protection is partial and does not guarantee immunity against emerging H5N1 variants, particularly those with significant antigenic drift from ancestral strains.

Additional ferret studies demonstrated that immunity from influenza A(H1N1)pdm09 more effectively reduced the replication and transmission of H5N1 virus than it did for H7N9 virus. This protection was supported by the presence of group 1 hemagglutinin stalk and N1 neuraminidase antibodies in pre-immune ferrets, suggesting that prior seasonal influenza exposure may confer some level of protection against influenza A(H5N1) clade 2.3.4.4b virus [43]. Studies of dairy farm workers in Michigan infected with H5N1 demonstrated the development of neutralizing antibodies post-infection [44].

Human antibody studies suggest that older individuals may have greater cross-reactive protection against H5N1. A study measuring H5N1 antibody responses in sera from 157 individuals born between 1927 and 2016 found that antibody titers to historical and recent H5N1 strains were highest in older individuals and correlated more strongly with birth year than with age, consistent with immune imprinting from exposure to other group 1 viruses (H1N1 and H2N2) during childhood [45].

This age-related immunity pattern suggests that younger populations might be more vulnerable in the event of an H5N1 pandemic, while older individuals might benefit from some degree of protection due to their prior exposure to other influenza strains. The study also measured H5N1 antibody responses in sera from 100 individuals before and after receiving an A/Vietnam/1203/2004 H5N1 vaccine. Both younger and older humans produced H5-reactive antibodies to the vaccine strain and to a contemporary clade 2.3.4.4b strain, with higher seroconversion rates in young children who had lower levels of antibodies before vaccination [44,45].

## 7. Treatment and Antiviral Options

Given the potential for H5N1 to cause severe disease in humans and a concerning report of an oseltamivir-resistant mutation in poultry from 8 of 45 farms in British Columbia, Canada [46], effective treatment options are essential. Recent research on antiviral efficacy shows that the selective MEK1/2 inhibitor zapnometinib (ZMN) significantly impairs viral replication across multiple HPAI strains, including H5N1 clade 2.3.4.4b. ZMN acts by causing nuclear retention of newly produced viral ribonucleoprotein complexes when the MEK/ERK/RSK1 kinase cascade is inhibited. Additionally, ZMN shows synergistic potential when used in combination with direct-acting antivirals like oseltamivir or baloxavir [47]. The detection of oseltamivir-resistant H5N1 variants in poultry highlights the importance of surveillance for antiviral resistance emergence, as widespread resistance could significantly limit treatment options during a pandemic.

A mouse study demonstrated that baloxavir, an FDA-approved influenza antiviral that works by inhibiting the virus’s cap-dependent endonuclease, improves disease outcomes (survival and reduced viral dissemination) over oseltamivir after lethal intranasal and ocular challenge with A(H5N1)-contaminated cow milk [47]. This finding is particularly relevant in the context of the dairy cattle outbreak, where milk contamination presents a novel route of potential human exposure.

Studies on the neuraminidase inhibitor susceptibility of H5N1 viruses isolated from humans in 2023–2024 found that most viruses remain susceptible to approved neuraminidase inhibitor antiviral drugs, including oseltamivir, zanamivir, peramivir, and laninamivir [48]. However, the emergence of drug-resistant viruses through spontaneous mutation or reassortment highlights the need for continued monitoring of antiviral susceptibility.

## 8. Vaccine Development

Several H5N1 vaccines have been approved for human use and are stockpiled in the US Government Strategic National Stockpile (SNS) to enhance pandemic preparedness. These vaccines, primarily based on inactivated or recombinant technologies, target specific H5N1 clades but may have limited efficacy against rapidly evolving variants due to antigenic drift. While they provide a critical foundation for outbreak response, their coverage against current and emerging H5N1 strains remains a challenge, necessitating advanced vaccine platforms with broader and more durable protection [49]. Additionally, an approved H5N1 vaccine is being administered to poultry and cattle in ongoing clinical trials, aiming to reduce viral transmission in agricultural settings, though its long-term impact on zoonotic spread is still under evaluation [50].

Progress in vaccine development is crucial for pandemic preparedness. Self-amplifying mRNA (sa-mRNA) vaccines have shown promise in preclinical models for influenza. Compared with N1-modified mRNA, sa-mRNA approaches demonstrated >100-fold greater transfection efficiency for multiple antigens with high durability for gene-of-interest production. In vivo immunogenicity studies showed that a 10-fold lower dose of sa-mRNA generated similar binding and neutralizing titers for avian pandemic influenza H5N1 in both mouse and rat models [51].

Sa-mRNA vaccine platforms also generated comparable or higher antigen-specific CD8 T-cell responses at 10-fold lower doses than conventional mRNA vaccines. Additionally, the lower doses of sa-mRNA generated a reduced elevation of reactogenic biomarkers while still generating similar or higher immunogenicity, suggesting the potential for dose sparing, improved durability, enhanced immunogenicity, and possibly reduced reactogenicity of the sa-mRNA platform for vaccine applications.

Adenoviral vector-based vaccine platforms expressing the H5N1 hemagglutinin stem region combined with autophagy-inducing peptide C5 conferred significant cross-protection against group 1 (H1N1 and H5N1) and group 2 (H3N2) influenza A viruses in mice, providing lower lung viral titers and reduced morbidity and mortality [52].

Another approach focuses on diversifying T-cell responses, particularly targeting conserved viral proteins such as the influenza A virus nucleoprotein (NP), which is highly conserved across IAV subtypes and less prone to antigenic drift than surface proteins like hemagglutinin. A mosaic nucleoprotein (MNP) vaccine that maximized the representation of 9-mer epitopes from thousands of NP sequences across human, swine, and avian IAVs has shown significant protection against both seasonal and pandemic influenza strains in mouse models by eliciting robust CD8+ T-cell responses that recognize conserved epitopes. This strategy leverages the NP’s stability to induce cross-protective immunity, reducing the need for annual vaccine updates and offering a promising tool for pandemic preparedness. In mouse models, the MNP vaccine reduced viral loads and morbidity against diverse IAV strains, suggesting potential broad protection against existing and emerging IAV strains, including H5N1 [53].

Research on broadly neutralizing antibodies has also made significant progress. Studies have identified monoclonal antibodies from plasmablasts following seasonal influenza vaccination that show exceptionally broad neuraminidase inhibition. One antibody, mAb-297, effectively protected mice against lethal doses of influenza A and B viruses, including H5N1, by targeting a common binding motif in the conserved neuraminidase active site with good tolerability [54]. These findings demonstrate that while B-cell responses against neuraminidase following conventional, egg-derived influenza vaccines are rare, inducing broadly protective neuraminidase antibodies through vaccination remains feasible and potentially safe.

## 9. Conclusions

The past decade has seen an alarming increase in H5N1 infections among a wide range of mammalian species worldwide [55,56], with a notable surge in cases across the Americas in recent years. This expanding host range, coupled with the acquisition of mammalian-adapting mutations and the emergence of new transmission pathways, underscores the pandemic potential of H5N1.

Key developments of concern include the 2024 U.S. cattle outbreaks featuring cow-to-cow transmission and milk contamination, evidence of mammal-to-mammal transmission in various species, foodborne infections, and neurological manifestations in infected mammals. These events highlight the ability of H5N1 to overcome species barriers, adapt to new hosts, and potentially lead to more severe disease outcomes.

Looking ahead, several priority areas warrant intensified focus. First, global surveillance infrastructure must be strengthened to enable rapid detection of emerging H5N1 variants with enhanced mammalian transmissibility. Second, the development of broadly protective, clade-agnostic vaccines using next-generation platforms such as self-amplifying mRNA and conserved epitope-targeting approaches should be accelerated. Third, the emergence of oseltamivir resistance underscores the need for diversified antiviral portfolios, including novel agents targeting host factors like MEK1/2 inhibitors. Finally, a coordinated One Health approach integrating human, animal, and environmental surveillance will be essential for early warning and rapid response to future spillover events.

While effective treatments and WHO-approved vaccines are available in case of a public health emergency, an H5N1 pandemic would still have significant health, social, and economic consequences worldwide. Therefore, continued surveillance, research, and preparedness efforts are crucial to better understand the factors driving the evolution and cross-species transmission of H5N1, assess the risks posed by emerging strains, and develop targeted interventions to mitigate the impact of a potential pandemic. International collaboration and a One Health approach encompassing human, animal, and environmental health will be key to addressing this complex global challenge.

## Data Availability

The original contributions presented in this study are included in the article. Further inquiries can be directed to the corresponding author.

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
