# Peer review of "Expanding Horizons: Host Range Evolution and Treatment Strategies for Highly Pathogenic Avian Influenza H5N1 and H7N9"

_viruses, 2025, doi:10.3390/v18010054_

Round 1

Reviewer 1 Report

Comments and Suggestions for Authors

This manuscript addresses a highly relevant topic for both human and animal public health. The authors present a valuable synthesis of current knowledge, and the following comments and suggestions are offered to strengthen the scientific rigor, clarity, and alignment with recent evidence.

Title The title “Expanding Horizons: Host Range Evolution and Treatment Strategies for Highly Pathogenic H5N1 Avian Influenza” specifies a focus on H5N1. However, the manuscript includes substantial discussion of H7N9. The authors should consider either redefining the title to reflect both subtypes or refocusing the content to align strictly with the stated title.

Abstract The abstract is clear and well-structured. Nevertheless, it should explicitly mention the predominant viral clade (2.3.4.4b) and highlight the potential risk of reassortment with human influenza viruses.

Introduction The classification of avian influenza viruses (AIVs) is appropriately presented. However, the section needs an updated references regarding the phylogenetic evolution of H5N1 in Latin America. Inclusion of recent data from regional outbreaks would strengthen the epidemiological context.

Epidemiology and Evolution The manuscript correctly addresses the expansion of H5N1 in mammalian hosts. However, it omits the 2023 outbreaks in South American sea lions (Peru and Chile), which are relevant to the discussion of cross-species transmission. Additionally, the human case reported in Ecuador should be cited with a formal bibliographic reference.

Molecular Aspects. The section provides a solid explanation of receptor binding affinity. It would be beneficial to add that dual affinity for α2,3- and α2,6-linked sialic acid receptors may facilitate co-infection with human influenza viruses, thereby increasing the risk of reassortment. Suggested placement: following the HA structural analysis.

Immune Response and Cross-Protection  The manuscript should clarify that cross-protection conferred by prior H1N1 exposure is partial and does not guarantee immunity against emerging H5N1 variants. Suggested placement: within the paragraph discussing human serological studies 

Treatment and Antivirals While the manuscript discusses zapnometinib and baloxavir, it does not address the emerging resistance to oseltamivir observed in avian and human H5N1 isolates. This issue should be discussed, ideally following the paragraph on neuraminidase inhibitor susceptibility 

Reviewer 2 Report

Comments and Suggestions for Authors

The review titled “Host Range Evolution and Treatment Strategies for Highly Pathogenic H5N1 Avian Influenza” is comprehensive, well-organized, and covers topics including virological characteristics, epidemiology, cross-species transmission, pathogenic mechanisms, immunology, treatment, and vaccine development. My review comments are provided below:

  1. Line 16: Please provide the most up-to-date data; there should be 71 human cases and 18 states.
  2. Line 21: Please include newly identified mammalian-adaptive mutations other than PB2-E627K and add the corresponding references.
  3. Lines 52-54: Since it is now 2025, it would be helpful if the authors could update the data to reflect the most recent information.
  4. The current content of this review is mostly descriptive and lacks critical or analytical discussion. The authors could appropriately provide scientific perspectives or evaluations, such as forward-looking discussions on global prevention and control strategies and policies, potential future trends in H5N1 outbreaks, and priority directions for vaccine and antiviral drug development.

Round 2

Reviewer 1 Report

Comments and Suggestions for Authors

The paragraph beginning with “The most dramatic evidence of H5N1’s expanding mammalian host range occurred during 2023 in South America…” is excessively long and conflates several distinct concepts (host range expansion, mass mortality, surveillance data, and population proportion). It is recommended to synthesize the key ideas so that the message is conveyed more clearly and directly.

Author Response

Reviewer Comment: The paragraph beginning with "The most dramatic evidence of H5N1's expanding mammalian host range occurred during 2023 in South America…" is excessively long and conflates several distinct concepts (host range expansion, mass mortality, surveillance data, and population proportion). It is recommended to synthesize the key ideas so that the message is conveyed more clearly and directly.

Response: We thank the reviewer for this constructive feedback. We have restructured the original single paragraph (256 words, 10 sentences) into three focused paragraphs with a logical progression:

1. Impact and Scale – Establishes the outbreak's significance and unprecedented mortality (24,000+ sea lions across five countries)
2. Clinical and Scientific Evidence – Presents surveillance data from Peru and Chile with clinical presentations and pathological findings
3. Geographic Spread and Transmission Mechanisms – Describes the 5,000 km Pacific and 2,800 km Atlantic spread patterns and transmission routes

This reorganization separates the previously conflated concepts while maintaining all scientific content and citations. The revised structure reduces cognitive load and allows readers to follow the progression from event description to evidence to mechanistic interpretation.